# Association of Vitamin D Prescribing and Clinical Outcomes in Adults Hospitalized with COVID-19

**DOI:** 10.3390/nu14153073

**Published:** 2022-07-26

**Authors:** Kathleen M. Fairfield, Kimberly A. Murray, A. Jerrod Anzalone, William Beasley, Maryam Khodaverdi, Sally L. Hodder, Jeremy Harper, Susan Santangelo, Clifford J. Rosen

**Affiliations:** 1MaineHealth Institute for Research, Portland, ME 04074, USA; kimberly.murray@mainehealth.org (K.A.M.); susan.santangelo@mainehealth.org (S.S.); clifford.rosen@mainehealth.org (C.J.R.); 2Department of Medicine, Tufts University School of Medicine, Boston, MA 02111, USA; 3Department of Neurological Sciences, University of Nebraska Medical Center, Omaha, NE 68198, USA; alfred.anzalone@unmc.edu; 4Biomedical and Behavioral Methodology Core, University of Oklahoma, Norman, OK 73019, USA; william-beasley@ouhsc.edu; 5West Virginia Clinical and Translational Science Institute, West Virginia University School of Medicine, Morgantown, WV 26506, USA; makhodaverdi@hsc.wvu.edu (M.K.); slhodder@hsc.wvu.edu (S.L.H.); 6Owl Health Works LLC, Indianapolis, IN 46278, USA; owlhealthworks@gmail.com

**Keywords:** COVID-19, vitamin D

## Abstract

It is unclear whether vitamin D benefits inpatients with COVID-19. **Objective:** To examine the relationship between vitamin D and COVID-19 outcomes. **Design:** Cohort study. **Setting:** National COVID Cohort Collaborative (N3C) database. **Patients:** 158,835 patients with confirmed COVID-19 and a sub-cohort with severe disease (n = 81,381) hospitalized between 1 January 2020 and 31 July 2021. **Methods:** We identified vitamin D prescribing using codes for vitamin D and its derivatives. We created a sub-cohort defined as having severe disease as those who required mechanical ventilation or extracorporeal membrane oxygenation (ECMO), had hospitalization >5 days, or hospitalization ending in death or hospice. Using logistic regression, we adjusted for age, sex, race, BMI, Charlson Comorbidity Index, and urban/rural residence, time period, and study site. Outcomes of interest were death or transfer to hospice, longer length of stay, and mechanical ventilation/ECMO. **Results:** Patients treated with vitamin D were older, had more comorbidities, and higher BMI compared with patients who did not receive vitamin D. Vitamin D treatment was associated with an increased odds of death or referral for hospice (adjusted odds ratio (AOR) 1.10: 95% CI 1.05–1.14), hospital stay >5 days (AOR 1.78: 95% CI 1.74–1.83), and increased odds of mechanical ventilation/ECMO (AOR 1.49: 95% CI 1.44–1.55). In the sub-cohort of severe COVID-19, vitamin D decreased the odds of death or hospice (AOR 0.90, 95% CI 0.86–0.94), but increased the odds of hospital stay longer >5 days (AOR 2.03, 95% CI 1.87–2.21) and mechanical ventilation/ECMO (AOR 1.16, 95% CI 1.12–1.21). **Limitations:** Our findings could reflect more aggressive treatment due to higher severity. **Conclusion:** Vitamin D treatment was associated with greater odds of extended hospitalization, mechanical ventilation/ECMO, and death or hospice referral.

## 1. Introduction

The COVID-19 pandemic has resulted in over 6.3 million deaths worldwide since March 2020 [1]. Although therapeutic advances are accelerating, efforts to understand the optimal management of people who are hospitalized with COVID-19 are critical to improving outcomes. Vitamin D is known to have complex effects on immunity, including activating innate immunity [2,3,4,5]. Vitamin D receptors are found on B cells, T cells, and antigen-presenting cells [6,7,8]. Vitamin D also modulates the respiratory epithelial and systemic response to viral infections [9,10]. It also reduces the overactive immune response, which may be relevant in the proinflammatory cytokine production and storm observed in severe infections [11,12,13] and acute respiratory distress syndrome (ARDS) [14]. A body of evidence from observational studies shows lower vitamin D levels in people with advanced age as well as chronic illnesses that are associated with poor COVID-19 outcomes, including chronic kidney disease and people with obesity [15,16,17]. However, this may simply be a marker of overall health and chronic disease burden. Efforts to supplement vitamin D for acutely ill patients have not demonstrated benefit in randomized trials conducted before the COVID-19 pandemic [18].

Studies examining the relationship between clinical outcomes from COVID-19 and vitamin D have produced varied results. In people with mild to moderate disease treated with high-dose vitamin D as outpatients, the time to clinical recovery was accelerated [19]. In an analysis of a large biobank in the UK, baseline vitamin D serum concentration was not associated with COVID-19 severity or mortality [20]. Among inpatients, a systematic review and meta-analysis suggested vitamin D supplementation is beneficial for reducing intensive care stays and mortality for people hospitalized with severe COVID-19 [21]. In contrast, a single high dose of vitamin D did not reduce length of stay in hospitalized COVID-19 patients with moderate to severe disease in a randomized trial [22]. Because there were no established practice guidelines for COVID-19 early in the pandemic, physicians may have prescribed vitamin D as part of a regimen aimed at the most likely mechanisms of disease progression, including cytokine storm and ARDS [12]. Inpatient physicians may also have been more inclined to prescribe vitamin D in patients experiencing more severe COVID-19 disease manifestations because of the relative safety of vitamin D, but we are unaware of any large-scale reports on vitamin D prescribing using national data.

The purpose of this analysis was to better understand the relationship between vitamin D given during hospitalization for COVID-19 and clinical outcomes in a national study according to underlying risk groups, including people with more severe disease and those at higher risk of poor outcomes.

## 2. Methods

We analyzed data from the National COVID Cohort Collaborative (N3C), a nationwide patient database created and managed by the National COVID Cohort Collaborative (N3C) [23]. The N3C is a partnership among NIH’s National Center for Advancing Translational Sciences (NCATS) programs including Clinical and Translational Science Awards (CTSA) Program hubs, the National Center for Data to Health (CD2H), and NIGMS-supported Institutional Development Award Networks for Clinical and Translational Research (IDeA-CTR), with overall stewardship by NCATS.

The N3C database is a centralized repository of electronic health record (EHR)-based clinical information on COVID-19 patients and controls submitted by 61 medical centers nationwide (as of 4 November 2021). Each study site provides demographic, medication, laboratory, diagnoses, and vital status data which is harmonized into the Observational Medical Outcomes Partnership (OMOP) data model. Information is available for COVID-19 encounters as well as a patient look-back data period to 1 January 2018 to provide information on pre-existing conditions and encounters. The N3C design, data collection, sampling approach, and data harmonization methods have been described previously [23,24].

This analysis included 158,835 patients diagnosed with severe acute respiratory syndrome associated with coronavirus-2 (SARS-CoV-2) who were hospitalized within 28 days of their COVID-19 diagnosis. We included patients 18 years of age and older, diagnosed between 1 January 2020 and 31 July 2021 (Figure 1). We defined COVID-19 using positive lab measurements (either polymerase chain reaction or antigen test), positive antibodies, or positive COVID-19 diagnosis (corresponding to U07.1 ICD10-CM code). We excluded patients from data submitters with lower-than-expected reporting for death and for drug exposure information (10 sites). As a result, we retained patients from 51 study sites.

The study outcomes included death or referral to hospice, length of stay longer than 5 days, and mechanical ventilation or extracorporeal membrane oxygenation (ECMO). We selected longer than 5 days as a criterion because the national data show that five to six days is the median length of stay for COVID-19 [25,26], and five days was the median in our study. We examined the relationship between vitamin D treatment during hospitalization and these three major outcomes. We categorized a patient as having received vitamin D if any dose of vitamin D was recorded, regardless of preparation. We were unable to reliably determine the dose, duration, or frequency due to source system information sparsity. Additionally, we were not able to fully capture vitamin D treatment prior to hospitalization because self-reporting of over-the-counter vitamin D is not available in the database. While information on patients prescribed vitamin D prior to hospitalization was available for those patients treated within the health systems submitting data to N3C, we do not have prescription data on patients outside of those reporting health systems.

The main independent variable of interest was vitamin D treatment during hospitalization. Descriptive analyses were performed to inform poor outcomes associated with SARS-CoV-2. The data were compared by outcomes using Student’s *t*-tests for continuous data and Pearson’s chi square tests were used as appropriate for categorical data. Logistic regression analyses were performed to identify the factors associated with death or referral to hospice, length of stay of 5 days or longer, and mechanical ventilation or ECMO.

The potential covariates were chosen a priori based on clinical knowledge and literature review. The models were adjusted for age, sex, race, year and calendar quarter of COVID-19 diagnosis, urban/rural residence, BMI, Charlson comorbidity score [27] (0, 1, 2, or 3 or more), and study site. The covariates and concept definitions are available in Appendix A. The analyses were conducted using SQL and R (v3.5.1) in the N3C data enclave (data release v52, 4 November 2021). We used the R package MatchIt (R Foundation for Statistical Computing, Vienna, Austria, v4.1.0) [28] for propensity scoring in separate sensitivity analyses using nearest neighbor, matching on vitamin D and using all covariates.

Because of hypotheses from the literature review and findings from the first phases of analysis that patients with more severe disease were more likely to receive vitamin D, we created a sub-cohort to examine relationships between vitamin D and outcomes in patients with severe COVID-19 disease (n = 81,381). Severe disease was defined as having experienced any one of the adverse outcomes under study: death or referral to hospice, length of stay longer than 5 days, or mechanical ventilation or ECMO.

## 3. Results

The characteristics of the full cohort included the following: 44% were aged 65 or older, 49% female, 52% white, 50% overweight or obese, and 35% had two or more comorbidities (Table 1). Eleven percent of the patients died or were referred to hospice (n = 16,838), 48% stayed longer than 5 days in the hospital (n = 75,751), and 13% required mechanical ventilation or ECMO (n = 20,417). Patients who were treated with vitamin D were older, had a greater number of comorbidities, and were slightly more overweight or obese compared with patients who did not receive vitamin D. In the full cohort of 158,835 patients, 28,993 (18%) received vitamin D during their hospital stay, while in the sub-cohort of 81,381 with severe disease, 18,132 (22%) received vitamin D. The majority of the patients who received vitamin D were treated within 5 days of hospitalization (88% among all patients, and 80% among patients with severe COVID-19).

We observed a higher percentage of each of the three adverse outcomes among patients receiving vitamin D in the full cohort (Table 2). In the sub-cohort of patients with severe COVID-19, a slightly lower percentage of people who received vitamin D died or were referred to hospice (20% vs. 21%). However, a higher percentage of those who received vitamin D had a length of stay greater than 5 days (96% vs. 92%), and required mechanical ventilation or ECMO (27% vs. 25%).

Table 3 shows the results from multivariable logistic regression models for vitamin D treatment compared with not receiving vitamin D during hospitalization for all three outcomes in the full cohort and those with severe disease. In the full cohort, vitamin D treatment during hospitalization was associated with an increased odds of death or referral for hospice (adjusted odds ratio (AOR) 1.10: 95% confidence interval (CI) 1.05–1.14), requiring a hospital stay longer than 5 days (AOR 1.78: 95% CI 1.74–1.83), and requiring mechanical ventilation or ECMO (AOR 1.49: 95% CI 1.44, 1.55). For patients with severe COVID-19, we observed a decreased odds of death or referral to hospice (AOR 0.90, 95% CI 0.86–0.94) and an increased odds of having a stay 5 days or longer (AOR 2.03, 95% CI 1.87–2.21) or requiring mechanical ventilation or ECMO (AOR 1.16, 95% CI 1.12–1.21) in those treated with vitamin D.

As a sensitivity analysis, we also conducted a propensity score analysis to assess the relationships between clinical factors and vitamin D prescribing. Our findings were not materially different using this approach.

We observed a marked variation in the use of vitamin D during hospitalization across the 51 study sites, from 3% to 40% (with one outlier, at 58%). Additional details from descriptive analyses and multivariable models are available in Appendix A.

## 4. Discussion

In this national cohort sample of people hospitalized with COVID-19, we observed an association between vitamin D administration and an increased risk of adverse outcomes including death, length of stay of more than 5 days, and the need for mechanical ventilation. However, among patients with more severe COVID-19, the receipt of vitamin D was associated with a lower risk of death or referral to hospice, despite the higher risk of requiring mechanical ventilation and longer length of stay. We also observed that 18% of inpatients received vitamin D during their hospital stay, although this varied substantially by reporting site, during a period when there was little evidence for its effect on COVID-19 outcomes.

Our findings are similar to another large cohort study showing no association between vitamin D status and length of stay or prognosis among patients with moderate to severe COVID-19 in adjusted analyses [29]. In contrast, a population-based cohort in Spain showed a modestly lower risk of severe disease and death in COVID-19 patients taking vitamin D [30]. Recent meta-analyses and systematic reviews of both randomized trials and non-randomized intervention studies showed a reduced risk of both the need for intensive care and mortality in people receiving supplementation in the hospital [21,31].

There are several possible explanations for our conflicting findings. Because vitamin D is associated with improved cellular immune function [2,32], the receipt of vitamin D may improve outcomes through improvements in the immune response. Additionally, or alternatively, because vitamin D may reduce cytokine storm [33], it may reduce the complications of COVID-19 in people with more severe disease [12]. It is possible that the most vulnerable and higher-risk patients with COVID-19 had lower vitamin D status at baseline, particularly since many are older and have comorbidities associated with poor health, and that vitamin D was beneficial in reducing death because of those relationships in the severely ill sub-cohort [34,35]. Our findings could also reflect the more aggressive treatment of people who were most ill from COVID-19, with vitamin D simply being part of a complex treatment plan. Although we adjusted for known risk factors for poor outcomes (e.g., age, obesity, and other comorbidities), residual confounding is always possible in cohort studies, and is, in fact, likely. Other clinical factors may have been driving prescribing practices, and although we use a propensity-matched approach for a sensitivity analysis, we have incomplete ascertainment of all clinical factors in this observational study.

This study had several limitations. We were unable to determine the dose, duration, or frequency of vitamin D, or the preparation for most patients, and therefore used “any” vitamin D receipt during the hospitalization as the exposure of interest. It is possible that the receipt of a specific dose or duration during a specific time window of the disease course could have been more effective for improving disease outcomes in selected patients with COVID-19. The baseline vitamin D status was not known, and it is also possible that people with poor outcomes also had suboptimal vitamin D status at baseline, therefore benefitting from vitamin D administration during hospitalization, while those with adequate vitamin D before hospitalization did not benefit from supplementation. As noted above, since this was an observational study, there may be unknown confounders or residual confounding even in the adjusted analyses.

In summary, we found a positive association between receiving vitamin D during hospitalization with COVID-19 and poor outcomes (death/hospice, extended length of stay, and mechanical ventilation or ECMO). Among patients with severe COVID-19 disease, we observed similar results except for death/hospice, which was inversely associated with vitamin D treatment. Large-scale clinical trials, already underway, are the best way to answer this question definitively, particularly with regard to nuanced differences in underlying vitamin D status, dose, timing, and duration. This work will inform additional randomized clinical trials and possibly clinical care pending those studies. In the interim, it may be prudent to only administer vitamin D to those with demonstrated deficiency.

## Figures and Tables

**Figure 1 nutrients-14-03073-f001:**
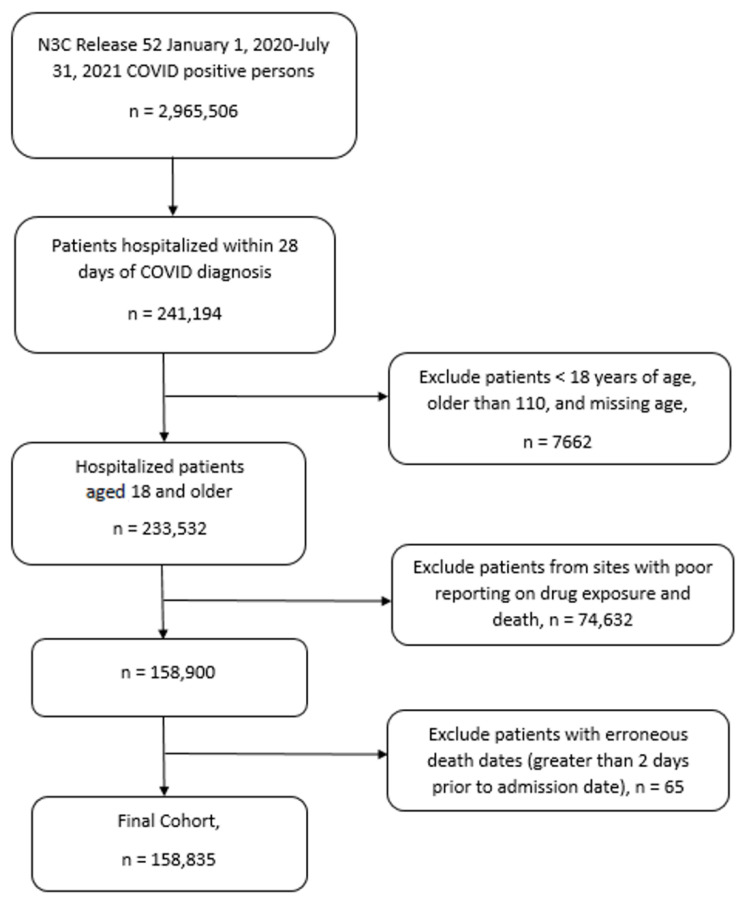
N3C Cohort Selection Consort Diagram.

**Table 1 nutrients-14-03073-t001:** Characteristics of inpatient COVID-19 cohort according to vitamin D treatment.

	All Patients		Patients with Severe COVID-19	
	No Vitamin D	Vitamin D	No Vitamin D	Vitamin D
	n = 129,842 (82%) ^1^	n = 28,993 (18%) ^1^	n = 63,249 (78%) ^1^	n = 18,132 (22%) ^1^
Age at Diagnosis				
18–29	11,153 (8.6%)	1012 (3.5%)	2538 (83%)	536 (17%)
30–49	28,434 (22%)	4374 (15%)	9982 (81%)	2394 (19%)
50–64	35,699 (27%)	8151 (28%)	17,884 (78%)	4936 (22%)
65–74	25,047 (19%)	6944 (24%)	14,293 (76%)	4486 (24%)
75 and older	29,509 (23%)	8512 (29%)	18,552 (76%)	5780 (24%)
Sex				
Female	63,349 (49%)	14,537 (50%)	27,233 (76%)	8664 (24%)
Male	65,643 (51%)	14,353 (50%)	35,585 (79%)	9394 (21%)
Unknown	850 (0.7%)	103 (0.4%)	431 (85%)	74 (15%)
Race				
White	5093 (3.9%)	1117 (3.9%)	31,891 (75%)	10,652 (25%)
Black or African American	25,252 (19%)	5459 (19%)	12,797 (79%)	3469 (21%)
Asian	4083 (3.1%)	808 (2.8%)	2633 (78%)	735 (22%)
Other	29,377 (23%)	4284 (15%)	2004 (77%)	613 (23%)
Unknown	66,037 (51%)	17,325 (60%)	13,924 (84%)	2663 (16%)
Ethnicity				
Hispanic or Latino	28,072 (22%)	5079 (18%)	12,705 (80%)	3136 (20%)
Not Hispanic or Latino	86,128 (66%)	21,546 (74%)	42,902 (76%)	13,510 (24%)
Unknown	15,642 (12%)	2368 (8.2%)	7642 (84%)	1486 (16%)
Quarter of Diagnosis				
2020 Q1	7033 (5.4%)	857 (3.0%)	4591 (87%)	699 (13%)
2020 Q2	28,417 (22%)	4522 (16%)	15,689 (83%)	3229 (17%)
2020 Q3	14,528 (11%)	2749 (9.5%)	6303 (79%)	1684 (21%)
2020 Q4	37,193 (29%)	9891 (34%)	17,521 (75%)	5963 (25%)
2021 Q1	27,857 (21%)	7322 (25%)	12,822 (74%)	4460 (26%)
2021 Q2	11,423 (8.8%)	2781 (9.6%)	4776 (75%)	1580 (25%)
2021 Q3 (July only)	3391 (2.6%)	871 (3.0%)	1547 (75%)	517 (25%)
RUCA Category: Patient Residence				
Urban	1812 (1.4%)	462 (1.6%)	41,349 (77%)	12,072 (23%)
Large rural	5996 (4.6%)	1973 (6.8%)	3298 (73%)	1233 (27%)
Small rural	3173 (2.4%)	837 (2.9%)	1802 (75%)	586 (25%)
Isolated	35,625 (27%)	6766 (23%)	1064 (77%)	318 (23%)
Unknown	83,236 (64%)	18,955 (65%)	15,736 (80%)	3923 (20%)
BMI				
Obese	17,040 (13%)	4159 (14%)	18,428 (75%)	6194 (25%)
Overweight	38,866 (30%)	9673 (33%)	11,606 (77%)	3493 (23%)
Normal weight	23,880 (18%)	5541 (19%)	9002 (77%)	2756 (23%)
Underweight	3069 (2.4%)	500 (1.7%)	1659 (83%)	351 (17%)
Unknown	46,987 (36%)	9120 (31%)	22,554 (81%)	5338 (19%)
Charlson Index Category				
0	42,385 (33%)	6786 (23%)	16,199 (80%)	3941 (20%)
1	15,617 (12%)	3350 (12%)	6999 (78%)	1919 (22%)
2	11,228 (8.6%)	2912 (10%)	5512 (76%)	1773 (24%)
3 or more	30,937 (24%)	10,137 (35%)	17,635 (73%)	6463 (27%)
Unknown	29675 (23%)	5808 (20%)	16904 (81%)	4036 (19%)

^1^ Statistic presented: n (%). Percent = row percent.

**Table 2 nutrients-14-03073-t002:** Outcomes by full cohort compared to patients with severe COVID-19 disease.

	All Patients with COVID-19			Patients with Severe COVID-19		
	No Vitamin D 129,842 (82%) ^1^	Vitamin D 28,993 (18%) ^1^	*p*-Value	No Vitamin D 63,249 (78%) ^1^	Vitamin D 18,132 (22%) ^1^	*p*-Value
Death/referral to hospice ^2^	13,185 (10%)	3653 (13%)	<0.001	13,185 (21%)	3653 (20%)	0.04
Length of stay > 5 days	58,372 (45%)	17,379 (60%)	<0.001	58,372 (92%)	17,379 (96%)	<0.001
Mechanical ventilation/ECMO	15,520 (12%)	4897 (17%)	<0.001	15,520 (25%)	4897 (27%)	<0.001

^1^ Statistic presented: n (%). Percent = column percent. ^2^ Death or referral to hospice during hospitalization.

**Table 3 nutrients-14-03073-t003:** Multivariable logistic regression models for outcomes associated with vitamin D receipt.

	All Patients with COVID-19		Patients with Severe COVID-19	
	Unadjusted OR (95% CI)	Adjusted OR * (95% CI)	Unadjusted OR (95% CI)	Adjusted OR * (95% CI)
Death/hospice	1.27 (1.23–1.33)	1.10 (1.05–1.14)	0.96 (0.92–1.00)	0.90 (0.86–0.94)
Length of stay > 5 days	1.83 (1.78–1.88)	1.78 (1.74–1.83)	1.93 (1.78–2.09)	2.02 (1.87–2.20)
Mechanical ventilation/ECMO	1.50 (1.45–1.55)	1.49 (1.44–1.55)	1.14 (1.10–1.18)	1.16 (1.12–1.21)

* Adjusted for age category, sex, race, time, (year/quarter), data partner, Charlson Comorbidity Index, vitamin D treatment, BMI, and RUCA code.

## Data Availability

All diagnostic, medication, procedure, and laboratory concepts used in this study are available in Appendix A. Raw code (R, SQL) is available upon request. N3C is a public resource maintained by NCATS to support COVID-19 research. To access patient-level data from the N3C consortium, institutions must have a signed Data Use Agreement executed with NCATS and investigators must complete mandatory training along with submitting a Data Use Request to N3C. Investigators can request access to the Enclave here (https://covid.cd2h.org/onboarding, accessed on 21 June 2022).

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
