# Peer review of "Association of Vitamin D Prescribing and Clinical Outcomes in Adults Hospitalized with COVID-19"

_nutrients, 2022, doi:10.3390/nu14153073_

Round 1
Reviewer 1 Report
Dear Editor and Authors,
Regarding the manuscript, the following observations could be mentioned:
1. Introduction chapter: References from the first paragraph; could be updated with data from the last 10 years; the last sentence in the second paragraph does not have a reference; the last sentence of the third paragraph could be moved to the Discussion chapter
2. Methods chapter: based on what criteria was the length of hospitalization higher than 5 days considered a severe adverse outcome? what were the criteria for starting or not vitamin D supplementation in these inpatients? what is Charlson comorbidity index (reference)?
3. Results chapter: where p-values calculated for Table 2? Data found in the general population is conflicting with data found in patients with severe COVID, can the authors discuss this finding?
4. Discussion chapter should be enriched with other relevant data related to vitamin D and COVID from the recent literature; here are a few examples: https://www.frontiersin.org/articles/10.3389/fphar.2022.836738/full ; https://pubmed.ncbi.nlm.nih.gov/35166850/ ; https://pubmed.ncbi.nlm.nih.gov/32869273/ ; https://www.ncbi.nlm.nih.gov/pmc/articles/PMC7385774/
The manuscript is interesting because of the conflicting findings, but it needs significant improvements in the Introduction and Discussion chapters.
Author Response
Editor and Authors,
Regarding the manuscript, the following observations could be mentioned:
- Introduction chapter: References from the first paragraph; could be updated with data from the last 10 years; the last sentence in the second paragraph does not have a reference; the last sentence of the third paragraph could be moved to the Discussion chapter
We thank the reviewer for this feedback and have updated several references accordingly. We did keep many of the classic basic science references in the manuscript, however, as they were foundation to our understanding of vitamin D and immunity.
We modified the final sentence of the 2nd paragraph as there are no publications on this issue that we could retrieve. In fact this current analysis may be one of the best pieces of evidence on this matter. We highlighted this important finding in the last sentence of the first paragraph of the Discussion.
We moved the last sentence of the 3rd paragraph to the end of the Discussion (final paragraph).
- Methods chapter: based on what criteria was the length of hospitalization higher than 5 days considered a severe adverse outcome? what were the criteria for starting or not vitamin D supplementation in these inpatients? what is Charlson comorbidity index (reference)?
We have added references and our rationale for selecting a LOS >5 days as one of our criteria for severe COVID-19.
Because this is an observational study, the physicians were deciding whether to prescribe vitamin D supplementation. We have not modified the manuscript on this point.
We added a sentinel references (Deyo et al) to the Methods section.
- Results chapter: where p-values calculated for Table 2? Data found in the general population is conflicting with data found in patients with severe COVID, can the authors discuss this finding?
We updated Table 2 to include the p values.
We believe that the 3rd paragraph of our discussion deals carefully with the conflicting findings and what might account for them. If the reviewer is looking for something more specific, we can try to address it. However, we feel we have described our findings cautiously and without over-interpretation.
- Discussion chapter should be enriched with other relevant data related to vitamin D and COVID from the recent literature; here are a few examples: https://www.frontiersin.org/articles/10.3389/fphar.2022.836738/full ; https://pubmed.ncbi.nlm.nih.gov/35166850/ ; https://pubmed.ncbi.nlm.nih.gov/32869273/ ; https://www.ncbi.nlm.nih.gov/pmc/articles/PMC7385774/
We appreciate the suggested references and have added them to the Discussion.
The manuscript is interesting because of the conflicting findings, but it needs significant improvements in the Introduction and Discussion chapters.
Submission Date
22 June 2022
Date of this review
29 Jun 2022 22:12:05
Reviewer 2 Report
The study aimed to assess the relationship between vitamin D and COVID-19 outcomes.
The main independent variable of interest was vitamin D treatment during hospitalization.
Patients who were treated with vitamin D were older, had a greater number of comorbidities, and were slightly more overweight or obese compared with patients who did not receive vitamin D
The authors releaved that Vitamin D treatment was associated with greater odds of extended hospitalization, mechanical ventilation/ECMO and death or hospice referral. Among patients with more severe COVID-19, receipt of vitamin D was associated with lower risk of death or referral to hospice, despite higher risk of requiring mechanical ventilation and longer length of stay.
COMMENTS
Numerous attempts in the form of systematic reviews and meta-analysis assess the potential role of vitamin D deficiency in COVID-19 infection, severity and mortality.
However majority of the reviews remained inconclusive and highlighted the need for more primary studies in the form of randomized controlled trials.
Furthemore studies analyzing the use of cholecalciferol or calcifediol supplementation to modify COVID-19 outcomes have offered inconclusive results, while some observational studies in hospitalized patients have shown reduced COVID-19 severity or mortality in patients supplemented with cholecalciferol or calcifediol . Cereda et al. described a trend to an increased mortality in patients supplemented with calcifediol .
The effects of cholecalciferol or calcifediol as a treatment for hospitalized COVID-19 patients have also been studied in three low-powered clinical trials, without observing any significant reduction in COVID-19 mortality.
The results of this study is very interesting despite some limitations that the authors well described.
The study adds controversial information about the issue.
More robust data from randomized controlled trials are needed to substantiate Vitamin D effects on clinical outcomes in Covid -19 infection.
The paper is well written and the authors did an appreciable work.
I have not major or minor remarks
Author Response
COMMENTS
Numerous attempts in the form of systematic reviews and meta-analysis assess the potential role of vitamin D deficiency in COVID-19 infection, severity and mortality.
However majority of the reviews remained inconclusive and highlighted the need for more primary studies in the form of randomized controlled trials.
Furthemore studies analyzing the use of cholecalciferol or calcifediol supplementation to modify COVID-19 outcomes have offered inconclusive results, while some observational studies in hospitalized patients have shown reduced COVID-19 severity or mortality in patients supplemented with cholecalciferol or calcifediol . Cereda et al. described a trend to an increased mortality in patients supplemented with calcifediol .
The effects of cholecalciferol or calcifediol as a treatment for hospitalized COVID-19 patients have also been studied in three low-powered clinical trials, without observing any significant reduction in COVID-19 mortality.
The results of this study is very interesting despite some limitations that the authors well described.
The study adds controversial information about the issue.
More robust data from randomized controlled trials are needed to substantiate Vitamin D effects on clinical outcomes in Covid -19 infection.
The paper is well written and the authors did an appreciable work.
I have not major or minor remarks
We appreciate the careful review of our manuscript and the comments above. We reviewed the manuscript by Cereda et al https://www.ncbi.nlm.nih.gov/pmc/articles/PMC7657015/ but did not add it to our Discussion as it referred to vitamin D status and supplementation prior to hospitalization, and did not apply directly to our Discussion of similar studies of vitamin D supplementation after hospital admission for COVID-19.
We absolutely agree that the randomized controlled trials are the key to understanding this issue, and we believe our Discussion adequately covers this.
Round 2
Reviewer 1 Report
I have no further comments.